# Consistent stoichiometric long-term relationships between nutrients and chlorophyll-a across shallow lakes

Daniel Graeber [1] ✉, Mark J. McCarthy[2], Tom Shatwell [3], Dietrich Borchardt [1], Erik Jeppesen[4,5,6,7,8], Martin Søndergaard[4,5], Torben L. Lauridsen[4,5] & Thomas A. Davidson [4]

Aquatic ecosystems are threatened by eutrophication from nutrient pollution. In lakes, eutrophication causes a plethora of deleterious effects, such as harmful algal blooms, fish kills and increased methane emissions. However, lake-specific responses to nutrient changes are highly variable, complicating eutrophication management. These lake-specific responses could result from short-term stochastic drivers overshadowing lake-independent, long-term relationships between phytoplankton and nutrients. Here, we show that strong stoichiometric long-term relationships exist between nutrients and chlorophyll $a$ (Chla) for 5-year simple moving averages (SMA, median $R^2 = 0.87$) along a gradient of total nitrogen to total phosphorus (TN:TP) ratios. These stoichiometric relationships are consistent across 159 shallow lakes (defined as average depth < 6 m) from a cross-continental, open-access database. We calculate 5-year SMA residuals to assess short-term variability and find substantial short-term Chla variation which is weakly related to nutrient concentrations (median $R^2 = 0.12$). With shallow lakes representing 89% of the world's lakes, the identified stoichiometric long-term relationships can globally improve quantitative nutrient management in both lakes and their catchments through a nutrient-ratio-based strategy.

Controlling eutrophication requires managing external total phosphorus (TP) and total nitrogen (TN) loads to lake ecosystems[1–4]. Universal patterns of N- or P-deficient phytoplankton growth have been proposed, with thresholds at molar TN:TP ratios < 20 for N deficiency and > 50 for P deficiency, primarily based on standardized laboratory measurements[5]. However, snapshot lake monitoring data have shown variable responses to changes in TN and TP concentrations at different TN:TP ratios, which was attributed to differences in lake or catchment characteristics[6], with apparent thresholds ranging from TN:TP = 13–22 for N deficiency and TN:TP = 51–62 for P deficiency[6]. Due to large variations in relationships between nutrients and chlorophyll $a$ (Chla) even among homogeneous lake sets, log-log transformations have been suggested for TP[7], and have been used to detect changes in apparent TN or TP control of phytoplankton biomass[8]. Furthermore, since measurements of TN and TP and phytoplankton biomass (e.g., Chla) are not independent, relationships are often considered

[1]Department Aquatic Ecosystem Analysis, Helmholtz-Centre for Environmental Research - UFZ, Magdeburg, Germany. [2]Chair of Hydrobiology & Fisheries, Estonian University of Life Sciences, Tartu, Estonia. [3]Department Lake Research, Helmholtz-Centre for Environmental Research - UFZ, Magdeburg, Germany. [4]Department of Ecoscience, and WATEC, C.F. Møllers Allé 3, Aarhus University, Aarhus, Denmark. [5]Sino-Danish Education and Research Centre, Beijing, China. [6]Limnology Laboratory, Department of Biological Sciences and Centre for Ecosystem Research and Implementation, Middle East Technical University, Ankara, Turkey. [7]Institute of Marine Sciences, Middle East Technical University, Mersin, Turkey. [8]Institute for Ecological and Pollution Control of Plateau Lakes, School of Ecology and Environmental Science, Yunnan University, Kunming, China. ✉e-mail: daniel.graeber@ufz.de

tautologous[7]. In terms of nutrient dynamics, many lake monitoring programs and datasets include only total nutrient concentrations and Chla, which do not include information on nutrient availabilities or lake internal nutrient dynamics[9,10]. Without mechanistic data describing internal nutrient dynamics and true availability, we are often left with total nutrient and Chla data with which to evaluate nutrient and phytoplankton relationships. We believe that by understanding the source of lake-specific, high variability of total nutrient – Chla relationships, and characteristics of the apparent tautology, global patterns of N and P relationships with phytoplankton growth in shallow lake ecosystems can be discerned.

Strayer et al.[11] suggested that long-term ecological signal analyses are suitable for three phenomena: slow processes, rare events, and subtle processes. Eutrophication may serve as a prime example of a subtle process, with links between Chla and nutrients becoming apparent only in the long term due to unpredictable year-to-year

variability obscuring subtle changes[11]. Random annual fluctuations in factors such as ice-free periods, changes in phytoplankton composition, phytoplankton pigment content, and top-down food chain effects contribute to substantial variation in phytoplankton responses to N or P changes[11–13]. By separating long-term signals from their short-term counterparts, a clearer relationship between phytoplankton biomass and nutrient concentrations should emerge (Fig. 1).

By integrating long-term signal analysis with a deeper understanding of the nutrient-Chla tautology, we may be able to better detect phytoplankton community growth patterns relative to TN and TP. Specifically, the tautology should only be valid during the growing season, when most of the total nutrient pool may be contained within phytoplankton biomass. Within the growing season, the tautology should not apply if a nutrient is present in excess, as illustrated by Lewis & Wurtsbaugh[7] for the TP-Chla relationship under P excess. We suggest that the tautology and nutrient accumulation have a

**A** Extraction of 5-year simple moving averages (SMA) and their residuals

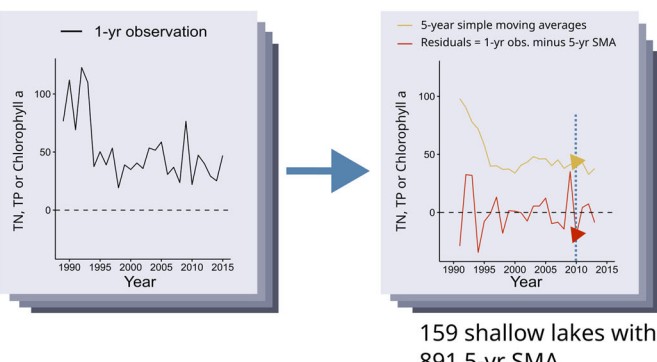

159 shallow lakes with 891 5-yr SMA

**B** Split into TN:TP range windows

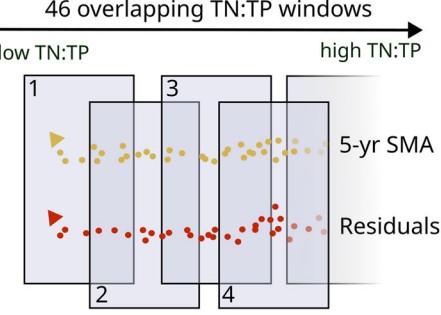

**C** Random sampling and correlation of Chla and nutrients within TN:TP windows

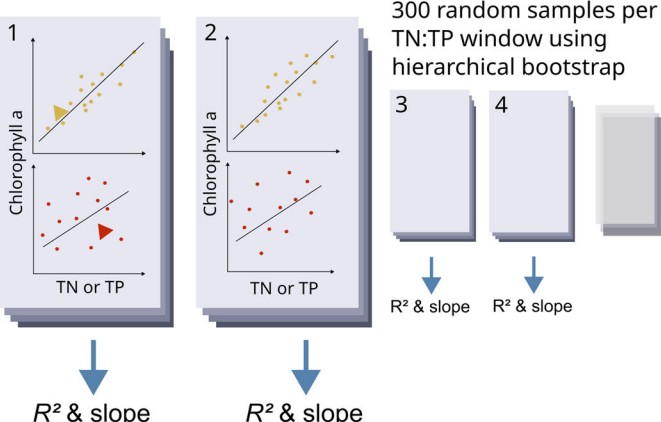

300 random samples per TN:TP window using hierarchical bootstrap

**D** 5-yr SMA, total nutrients versus Chla along TN:TP

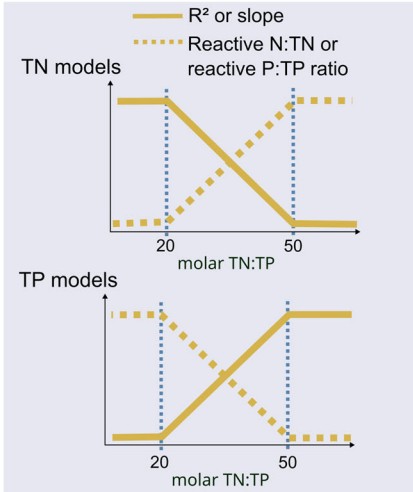

**Fig. 1 | Conceptual diagram of the data analyses performed and their hypothesised results.** We used growing season averages of shallow lake time series to calculate 5-year simple moving averages (SMA), which act as low-pass filter to extract long-term variation. We used the residuals of the 5-year SMA, calculated as the growing season observation minus the SMA for the same year, as high-pass filter to reveal short-term variation (**A**)[12]. We evaluated different SMA lengths and found 5 years to be ideal (see methods and SI for details). We categorized the data into overlapping TN:TP "windows" to assess the link between TN, TP, and Chla within different ranges of TN:TP (**B**). Here, we used the natural logarithm of TN:TP ratios because ratios follow a log-normal distribution[59]. For each TN:TP window,

the data were randomly sampled 300 times using a hierarchical bootstrap procedure to preclude temporal dependence of the sampled data[62]. For each random sample, we calculated generalized linear models with gamma distribution (5-year SMA) or linear models (SMA residuals) for the relationship between Chla and TP and/or TN, and extracted $R^2$ and model coefficients (**C**). Model slopes and $R^2$ are shown in the main text, see Methods and SI for details on the bootstrap and other model coefficients. We hypothesize high TP model slopes, high $R^2$ and low concentrations of reactive pelagic P in the form of soluble reactive phosphorus (SRP) at higher TN:TP, and high TN model slopes, high $R^2$ and low concentrations of reactive pelagic N as nitrate-N at lower TN:TP (**D**).

stoichiometric variant, where dissolved nutrients can accumulate when phytoplankton assimilation of a dissolved nutrient is constrained by the availability of another nutrient (Fig. 1). In this case, the nutrient in relative excess may accumulate in the water column in dissolved, reactive form[5,6]. This notion agrees with previous ecological models[14,15], which have proposed stoichiometric limits to phytoplankton N and P cell contents depending on relative water N and P concentrations, with accumulation of reactive nutrients if these nutrients are available in excess. Furthermore, this approach assumes that concentration measurements of the reactive nutrient forms can capture the true nutrient bioavailability in the system to a large part. If the proposed patterns of relative N and P accumulation and availability are lake-independent, universal relationships between TN, TP, reactive nutrient forms and Chla should emerge along a gradient of TN:TP ratios in shallow lakes.

We propose that long-term signals capture the majority of statistical dependence of Chla concentrations on TN and TP concentrations, and that the strength of the relationships and model coefficients are universal along a TN:TP gradient for a large, global shallow lake dataset (Fig. 1). If most of the information about relationships between nutrients and Chla is contained within the 5-year simple moving averages (SMA), then year-to-year residual variation should not reveal systematic changes in model slope or high $R^2$[12]. Long-term relationships between Chla and TN or TP concentration during the growing season should exhibit a consistent, lake-independent steeper slope (Fig. 1)[16]. Additionally, the long-term pattern should indicate an increased proportion of inorganic, reactive forms of N and P during the growing season, when there is N and/or P excess for phytoplankton growth at specific TN:TP ratios (Fig. 1). Of the inorganic, reactive N forms, ammonium-N is most energetically favorable for algal growth[17,18], but concentration measurements are scarce and unreliable due to its bioreactivity[19]. Instead, we chose nitrate-N, as it is less energetically favorable, and thus more prone to accumulate at measurable concentrations, and more likely to accumulate over long-term periods[20,21].

We use lakes with a cut-off of 6 m average depth and without any further data selection. By doing so, our data analysis is representative of the vast majority of the world's lakes; of the 1.4 million lakes in the global HydroLAKES database[22], 89% are shallower than 6 m. We used this cutoff to assess shallow, often polymictic lakes where water samples of nutrients and Chla are likely to be representative of the entire water column[23].

In this work, we show robust, ubiquitous stoichiometric long-term relationships between nutrients and Chla in shallow lakes, supporting the current view that nuanced dual nutrient control based on nutrient ratios is required for efficiently managing lake eutrophication[2,6,24]. For such strategies to succeed, it is crucial to manage both N and P across the entire catchment area, adopting a comprehensive and integrated perspective on nutrient loads and ratios[25–27]. However, further investigation is necessary to confirm whether the observed stoichiometric patterns represent a consistent, causal relationship between nutrient levels and phytoplankton growth over the long term. If future studies validate our findings, the identified stoichiometric relationships will enable managers to accurately predict the outcomes of long-term eutrophication management, based on TN:TP ratios and concentrations.

## Results and discussion

### Nutrient - chlorophyll a links for long-term averages

When combining TN and TP concentrations as predictor variables in additive models, the long-term, 5-year SMAs showed the underlying relationships between TN, TP and Chla with a high median $R^2 = 0.87$ (0.69–0.96 95% highest-density interval (HDI)) for the additive linear models (Fig. 2C, see SI for Akaike's Information Criterion supporting this statement). Although uncertainty of the additive models increased

at TN:TP < 20, we still found a median $R^2 = 0.72$ (0.47–0.89 HDI) (Fig. 2C). The higher uncertainty around $R^2$ is shared between all models for TN:TP < 20 (Fig. 2A–C) and is probably due to the smaller number of observations in this range (16–64 observations per model compared to 18–158 observations per model for TN:TP > 20, see also SI).

The slopes of the additive models revealed a tipping point behavior with change of TN:TP (Fig. 3B, D). Here, the TP slopes of the additive models (Fig. 3B) were maximal at TN:TP > 50 (median = 381.5, 130.8–524.5 HDI), with a decline at TN:TP < 50, and TP slopes partially crossing zero at TN:TP < 20 (median = 43.4, −54–180.2 HDI). Furthermore, the additive model slopes for TN revealed a clear tipping point behavior (Fig. 3D), with the highest values at TN:TP < 20 (median = 54.9, 28.7–86.6 HDI), a gradual decrease for TN:TP of 20–50, and model slopes consistently near zero at TN:TP > 50 (median = 5.8, 1.4–11.3 HDI). Previous approaches have not been able to assess whether N or P, both, or neither nutrient is closely linked to Chla in the range between TN:TP of 20–50[5,6]. In contrast, our results show empirical evidence of ubiquitous dual-nutrient links to Chla for the 5-year SMAs within this TN:TP range (Fig. 3). Furthermore, the majority of our 5-year SMA observations (60%) (Fig. 3) were between TN:TP = 20–50, which suggests that dual-nutrient links to Chla are the most common scenario in lakes <6 m average depth, which is in agreement with a global cross-ecosystem analysis of relative nutrient limitation[28].

When only including TP concentration as the predictor variable, the long-term relationships of TP - Chla also agree with our assumptions (depicted conceptually in Fig. 1D) and previous literature[8]. For TN:TP > 30, TP - Chla models produced a median $R^2 = 0.87$ (0.8 − 0.94 HDI), which decreased to a median $R^2 = 0.5$ with high uncertainty (0.08–0.76 HDI) for TN:TP < 20 (Fig. 2A). Here, the slope change of the TP - Chla models revealed a tipping point behavior, where the single-nutrient TP model slopes were stable at TN:TP > 30, and decreased below that, with slopes of a portion of the model solutions crossing zero at TN:TP < 15 (Fig. 3A). The implication of relative TP deficiency or excess from our model estimates was also supported by the accumulation of SRP. Specifically, median SRP concentrations were 0.17 mg L$^{-1}$ for TN:TP < 20 and 0.01 mg L$^{-1}$ for TN:TP > 20 (Fig. 4A).

When only including TN concentration as the predictor variable for TN - Chla models, we observed unexpected, yet potentially revealing patterns. The $R^2$ of the TN-only models (Fig. 2B) was high even at TN:TP > 50, which contrasts with the expected relative N excess and breakdown of $R^2$ (Fig. 1D). Furthermore, the TN-only model slopes revealed no tipping point behavior indicative of change in N excess, but a continuous and highly consistent slope decrease with increasing TN:TP (Fig. 3C). We also found a dichotomy of nitrate-N concentrations at TN:TP > 50, with a small subset of data revealing high nitrate-N concentrations of more than 1 mg L$^{-1}$ (13% of the samples), but most samples showing nitrate-N concentrations < 0.1 mg L$^{-1}$ (77%, Fig. 4B). Based on this it seems that, in the majority of lakes, denitrification was efficient at removing nitrate-N[29,30], and was responsible for the low correlation between TN and nitrate-N (see SI for a TN - nitrate-N scatterplot), except for very high TN concentrations, as was shown before for nutrient-rich lakes[31]. For the data with low nitrate-N concentrations, we also detected lower Chla concentrations, indicating that potentially reactive N was depleted even at high TN:TP (Fig. 4B). Thus, the data suggests that N was not as available to phytoplankton at high TN:TP (Fig. 2B, Fig. 3C) as P seems to have been at low TN:TP (Fig. 2A, Fig. 3A).

Based on the empirical evidence, we propose that non-reactive or less reactive N pools must contain the N not accessible for phytoplankton growth at high TN:TP. Living or dead particulate organic matter is not a likely storage pool, as typical ranges of phytoplankton[32] and bacterial[33] cell N: P ratios are insufficient to account for such high TN:TP ratios. In contrast, dissolved organic matter, particularly dissolved proteinaceous matter, is a likely candidate. In a study of four

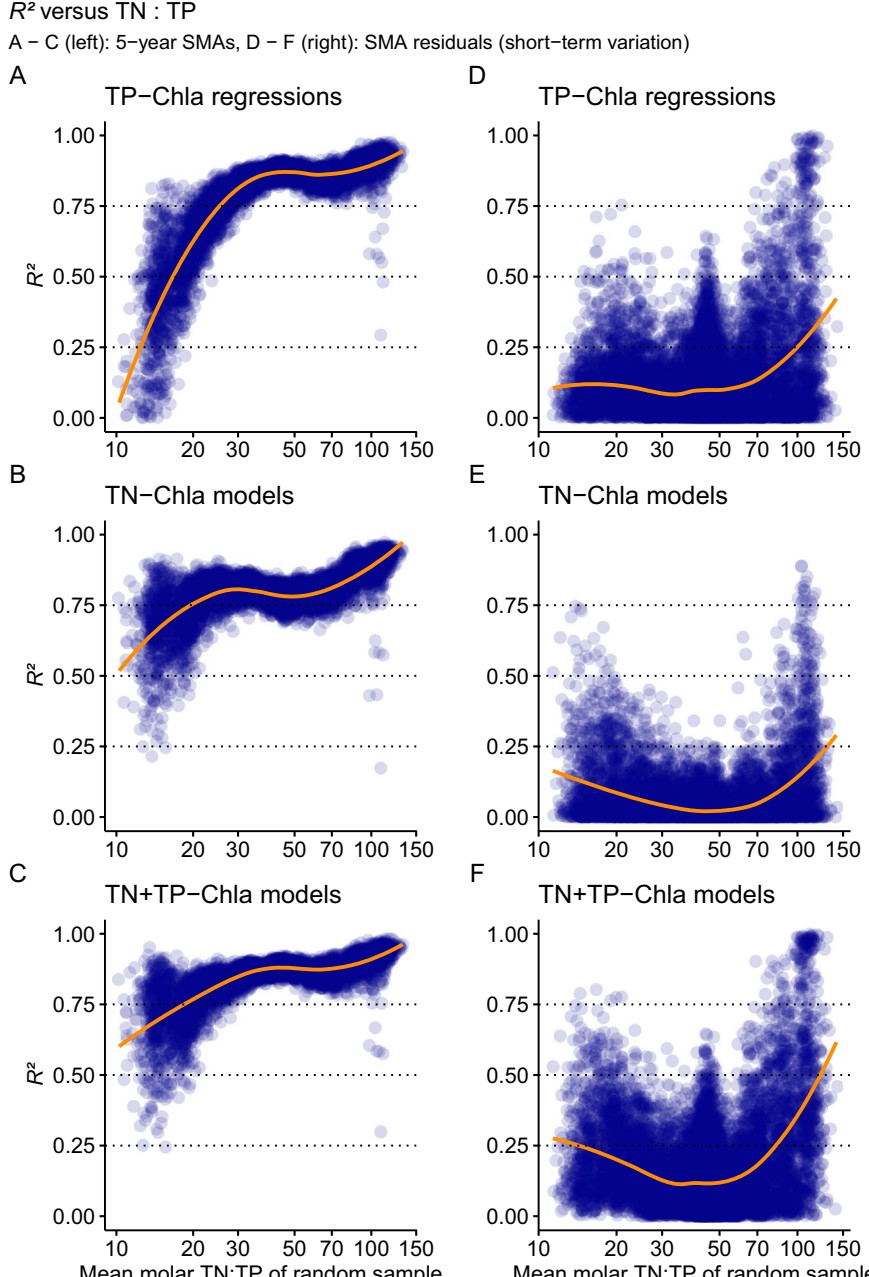

**Fig. 2 | Explained variance R² of the generalised linear models.** The R² values are separated for long-term variation in the shallow lakes based on 5-year simple moving averages (SMA) (**A–C**) and short-term variation based on SMA residuals (**D–F**). Shown are results from generalised linear models with gamma distributions for 5-year SMAs and linear models with normal distributions for SMA residuals for total phosphorus (TP, mg/L, panels **A**, **D**), total nitrogen (TN, mg/L, panels **B**, **E**), or additive models of TP and TN (panels **C**, **F**) and chlorophyll a (Chla, μg/L). These are plotted against the mean molar TN:TP of each randomly sampled dataset (see Fig. 1 for data processing steps). The less transparent the points, the more overlapping solutions for R² were found by the bootstrap procedure (indicating the error of R²). The orange line is the mean response based on a LOESS function. $N = 9194$ samples for the 5-year SMAs because not all models converged (**A–C**) and $n = 13800$ samples for the SMA residuals (**D–F**).

lakes, dissolved proteins were selectively preserved over decadal to centennial timescales potentially due to bacterial and photochemical degradation[34]. Moreover, in a long-term study of Flathead Lake, high pelagic molar TN:TP ratios (80 – 90) were best explained by long-term N conservation in the dissolved organic N pool[35]. In our case, TN concentrations were high at TN:TP > 50, and were maximal at TN:TP > 100 (see SI, section 9, for details). Hence, refractory dissolved organic N may be a considerable, yet largely overlooked, N storage pool in lakes. Based on this evidence, we hypothesise two alternative pathways for the N fate in high TN:TP lakes: (i) either accumulation of excess N as nitrate-N in a minority of lakes, or (ii) removal and storage

in a coupled denitrification-dissolved organic N pathway, with accumulation of refractory dissolved organic N, likely dissolved proteins, in the majority of lakes (Fig. 3B). Such a coupled storage pathway resembles the microbial carbon pump concept in oceans[36] and soils[37]. Furthermore, as organic carbon availability often controls both denitrification and bacterial N assimilation[26,38,39], dissolved organic carbon:N ratios could be crucial for delineating the dominant long-term N pathway.

Another aspect important for the link between TN:TP and N depletion is the potential importance of N fixation by cyanobacteria for providing N to phytoplankton growth. An earlier multi-lake study

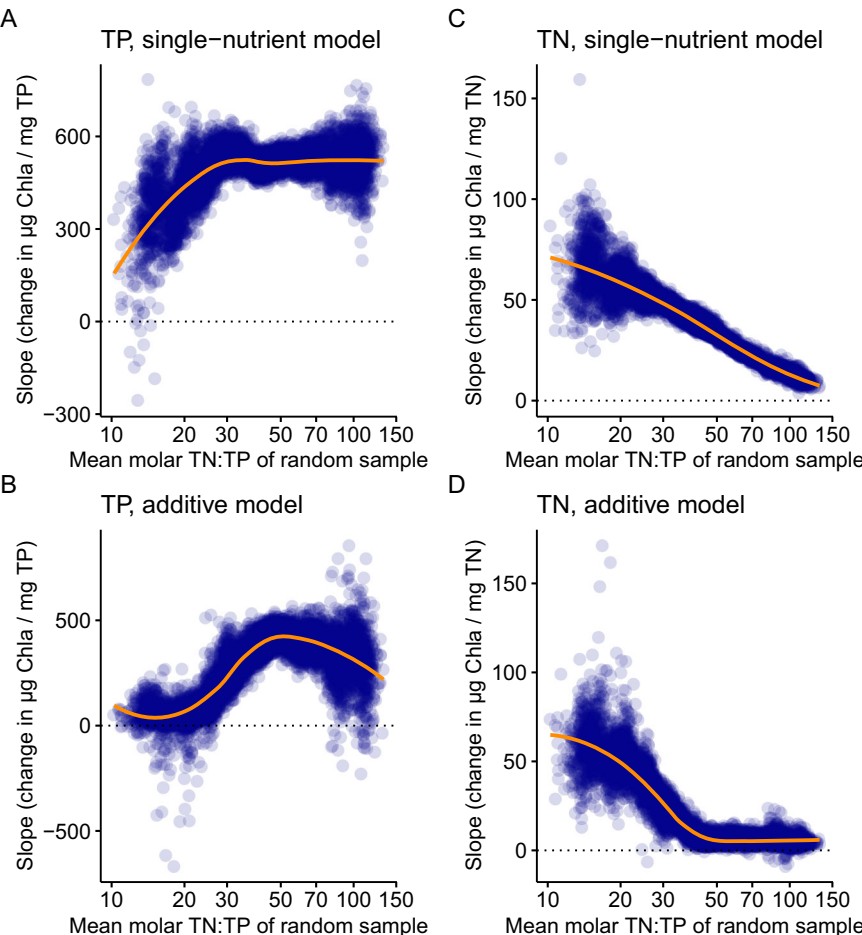

5−year SMA model slopes
Single−nutrient and additive models versus TN : TP

**Fig. 3 | Slopes of the generalised linear models.** Slope values are shown for single-nutrient and additive models for long-term variation in the shallow lakes based on 5-year simple moving averages (SMA) for TP (**A**, **B**) and TN (**C**, **D**) versus mean molar TN:TP of each randomly sampled dataset. Shown are results from generalised linear models with gamma distribution of Chla for 5-year SMAs. The less transparent the points, the more overlapping solutions were found for the slopes by the bootstrap procedure (indicating the error of the slopes). The orange line is the mean response based on a LOESS function. $N$ = 9194 samples.

revealed relatively higher importance of denitrification over atmospheric N fixation in the majority of studied lakes[24]. In our study, high relative N fixation rates would lead to long-term N excess even at low TN:TP ratios[40,41], causing consistently low $R^2$ and no systematic slope changes across the entire TN:TP range, which we did not observe (Fig. 2A, C). Alternatively, constantly high rates of N fixation relative to denitrification would push TN:TP consistently towards N excess at higher values of approximately TN:TP > 50. Again, we did not observe this, as our study revealed that the majority of TN:TP were between 20 and 50 (60% of the observations), supporting the prevalence of denitrification.

**Comparison of nutrient-chlorophyll a links in long-term and short-term data**
By extracting the 5-year SMAs and using a bootstrap procedure, we revealed a robust, ubiquitous long-term response across 159 shallow lakes, suggesting a common long-term stoichiometric relationship between nutrients and Chla, irrespective of differences in nutrient concentrations, lake characteristics, or catchment type. In contrast, the short-term variation contained within the SMA residuals showed low correlation coefficients (median $R^2$ = 0.12, 0−0.46 HDI, Fig. 2D−F). Therefore, the persistence of strong phytoplankton-nutrient relationships only in the 5-year SMAs (Fig. 2A−C) supports early ideas that

nutrient control of eutrophication at the ecosystem scale is a gradual process[11,42]. This increased robustness of the link between nutrients and phytoplankton also removes the need to log transform nutrient concentrations and phytoplankton biomass concentration variables, an approach which has been used to detect changes in nutrient-phytoplankton links when using snapshot samples or shorter-term means[8]. However, this long-term link may be surprising as phytoplankton has generation times of days to weeks. That the effect of eutrophication is longer lasting than the generation time of phytoplankton is reminiscent of earlier research, where a a delayed response of species to disturbances over several generations has been proposed to mimic the true disturbance impact[43]. Therefore, multi-year averages may reveal delayed effects of disturbances, but also effects of slowly changing drivers such as climate[12] or terrestrial catchment changes.

Of all the lakes contained in the large open-access global database, only 159 lakes in the temperate climate zone had sufficient long-term data to assess long-term phytoplankton-nutrient relationships, limiting our conclusions for Arctic or tropical shallow lakes. For lakes outside the temperate zone, we assume similar stoichiometric relationships due to similarities in algal physiology. However, marine studies suggest shifted thresholds due to differences in phytoplankton community composition in different climate zones[44], and similar effects may apply in lakes.

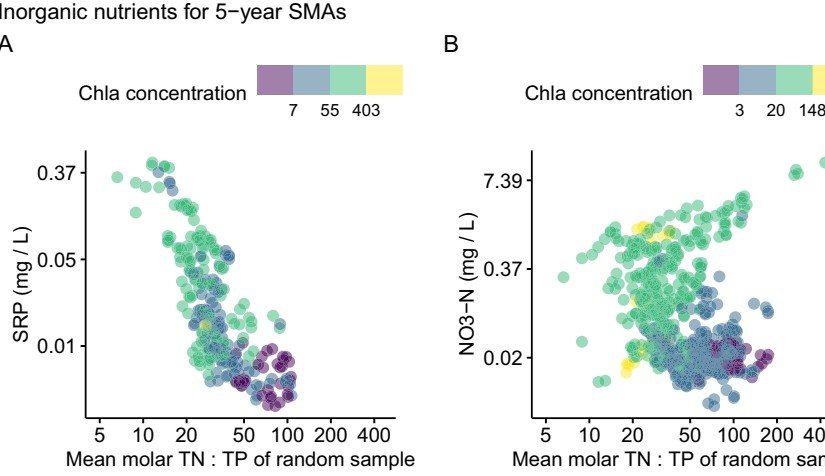

**Fig. 4 | Dissolved inorganic nutrient concentrations versus TN:TP ratios of the shallow lakes.** Shown are soluble reactive phosphorus (SRP) (**A**) and nitrate-N (NO3-N) (**B**) concentrations of each 5-year simple moving average (SMA) versus the molar TN:TP ratio of the same lake and 5-year SMA. Data are based on the same time series as the other data, but SRP and nitrate-N data were not available for every TN and TP observation. Therefore, panel **A** includes 278 observations from 37 shallow lakes, and panel **B** includes 620 observations from 138 shallow lakes. The axes have been log-transformed since TN:TP ratios followed a log-normal distribution, and to make small SRP and nitrate-N concentrations visible.

Considerable variations in Chla, TN, and TP concentrations were present in residual short-term data. These short-term residual variations were approximately half has high as for the 5-year SMAs. Specifically, the variation of the 5-year SMAs and their residuals can be expressed as the 95% range of their values using the HDI. For the 5-year SMAs, the HDI range of Chla was 133 μg L$^{-1}$. For the short-term Chla data, the HDI range was 73.7 μg L$^{-1}$. Similarly, for the 5-year SMAs versus the SMA residuals, we found HDI ranges of 2.7 versus 1.1 mg L$^{-1}$, and 0.27 versus 0.12 mg L$^{-1}$ for TN and TP, respectively.

This remaining high short-term variability in Chla may be related to variations in abiotic parameters controlling phytoplankton biomass, such as different N and P fractions, and to food web effects[12]. For example, fish predation on cladocerans can vary widely between years due to differences in environmental factors and strongly affect Chla concentrations[45]. Another source of short-term variation could be the Chla content of phytoplankton, which can vary widely within and between species[13]. Furthermore, links between variability in external and internal loading of N and P, and their specific bioavailable fractions need to be elucidated to better understand short-term variability. Here, not only the well-established internal P loading mechanisms, but also short-term pulses of internal N loading, both from sediments[10,46] and the water column[9,47], may play an important role in causing and maintaining eutrophication. Such pulses may be overlooked by infrequent spot sampling of TN and TP. Furthermore, simple linear regressions may be incapable of revealing short-term eutrophication responses to external drivers. Instead, these responses may emerge in short-term fluctuation amplitude, covariance, synchronicity, and frequency of the variability of nutrients and phytoplankton, analogous to findings for short-term variation of *Daphnia* and its food sources in lakes[48]. We propose that disentangling the sources and mechanisms of the revealed short-term variation requires high-frequency, detailed measurements of true nutrient availabilities, phytoplankton biomass, and food web structures, linked to mechanistic nutrient flux measurements[24].

Beyond lakes, many other ecological responses may have two temporal components: a short-term component linked to microbial food webs or stochastic short-term changes, and a long-term component linked to factors with variability following a longer periodicity. This concept has been suspected for decades[11,43] – and many researchers instinctively use long-term trends to remove short-term variation – but we found few rigorous, systematic investigations of multi-year monitoring or experiments. One notable exception is a recent study by Cusser et al.[49], who found that 12 long-term freshwater and terrestrial ecosystem experiments on plant and animal composition exhibited consistent reposes to experimental manipulation only over periods typically exceeding 10 years. With increasing access to high-resolution long-term ecological data, our study and that of Cusser et al.[49] may inspire further research using similar approaches to temporal signal extraction and random time series sampling.

## Usage of long-term stoichiometric relationships in lake and catchment management

The long-term relationships between nutrients and Chla revealed in this study provide empirical evidence for the often debated long-term stability of dual nutrient control of phytoplankton biomass[40], and strongly suggest that recent considerations of dual nutrient control[2,16,50] are the best approach to lake management for the majority of shallow lakes. However, dual nutrient control is not easily implemented, as managing N and P input to shallow lakes is challenging.

Many shallow lakes are supplied by surface waters and are part of extensive freshwater networks, where the majority of external N and P originates from hydrological catchment sources. For P, source control has been successful in reducing P loading from point sources[16,40,51]. However, diffuse N and P source control often demonstrates limited success[25,51,52]. As a result, well-designed nature-based solutions are necessary to physically and biologically retain N and P within soils, riparian zones, wetlands, ponds, and stream networks before these enter lakes, while providing beneficial secondary effects (e.g. on biodiversity) and avoiding deleterious effects (e.g. increased greenhouse-gas emissions and pollution swapping)[53,54]. Furthermore, our results may also be applicable for deeper lakes, as we find highly similar results when we do not only include shallow lakes in the data analysis (see SI for an analysis without depth cut-off). Yet, deeper lakes only comprise a small part of the dataset, and we recommend conducting a separate analysis for a dataset consisting of only deeper lakes to assess whether similar patterns appear.

Coupling monitoring, ecosystem models, and experiments may provide additional mechanistic evidence that the stoichiometric empirical patterns found here represent a causal link between nutrients and phytoplankton growth. If our findings are supported by further studies, the stoichiometric relationships found in this study will

allow managers to make long-term predictions of eutrophication management success based on TN:TP ratios and their concentrations.

## Methods

### Data collection
We collected data from the largest currently available open-access global lake database[55], to which we added data provided by the Danish Centre for Environment at the Aarhus University in an open-access database (https://odaforalle.au.dk, downloaded in February 2022). We collected average depth and concentrations of Chla, TN, and TP. We performed all data analyses and statistics in R, version 4.3[56].

We included all lakes with an average depth <6 m. With this cutoff, our data analysis represents the vast majority of the world's lakes. Of the 1427688 lakes in the global HydroLAKES database[22], 89% are shallower than 6 m. We used the 6-m cutoff, since the probability of stratification is <0.5 for an average lake depth of <6 m[23]. Thus, this cut-off represents lakes with a higher probability of mixing than that of stratification, resulting in the fact that water samples of nutrients and Chla should more likely be representative of the entire water column. Not all lakes in the global nutrient dataset had mean depth values[55], so we added mean depths from the HydroLAKES database to the global nutrient data time series if not already available. This approach was based on geographical location of the lake using QGIS[57] and the unique lake identifier in the global nutrient database[55]. The Danish database included depth values for all lakes, and we excluded saline or brackish waters with more than 2000 µs cm$^{-1}$. No other data filtering, outlier detection, or exclusion was performed.

The global nutrient database did not contain nitrate-N or SRP concentration data. Therefore, we extracted these data from the LAGOS-NE database using the LAGOSNE R package (version 2.0.2, database version 1.087.3) in R (version 4.3.0), which contains lake data from the northeastern USA[58]. The lakes from LAGOS-NE are included in the global nutrient database[55]. The Danish database contained nitrate-N and SRP concentration data. Again, we kept only lakes <6 m mean depth, and no other data filtering was applied.

### Calculation of growing season means and simple moving averages
For each lake and year, we calculated growing season means for TN, TP, nitrate-N, SRP, and Chla if three or more observations were available within this period in a given year. For the Northern Hemisphere, we used May to September as the growing season, and for the Southern Hemisphere, we used November to March. No lakes from the tropics were included in the final dataset due to a lack of long-term data series.

After assessing the ideal SMA length by using simulations and the available time series data, we found that 5-year SMAs remove most of the short-term variability while keeping long-term information intact and keeping a high number of lake time series in the dataset (please see the SI for the detailed approach). Based on the growing season means, we calculated all available 5-year SMAs for each lake. We then calculated SMA residuals as the growing season mean minus the 5-year SMA for the same year and lake. Subsequently, we used the SMA to split the lake time series into high-frequency and low-frequency data. This approach has been used previously for a single lake phytoplankton time series[12], and we extended this approach to multiple lake time series. Within the SI, we provide evidence for the usefulness of this approach to separate short- and long-term signals in time series data.

### Separation into different TN:TP ratio ranges
To assess the effects of TN or TP on associations with Chla at different TN:TP, we separated the 5-year SMA data into TN:TP 'windows'. We generated 46 TN:TP windows within a range of molar TN:TP between 1 and 1808. Nutrient ratios follow a log-normal distribution[59], so the TN:TP windows were log-transformed (natural logarithm (ln)). The windows ranged from ln TN:TP = 1 to 7.5, with a window width of ln TN:TP = 3 and an increment of ln TN:TP = 0.1. The windows overlapped; for example, the first window had a range of ln TN:TP from 1 to 3, and the second window ranged from 1.1 to 3.1, and so on. We assigned the 5-year SMA and SMA residuals to their TN:TP windows (usually more than one, since the TN:TP windows overlapped). The SMA residuals do not contain absolute TN:TP ratio data, so they were assigned based on TN:TP ratio data for the 5-year SMA for the same year and lake.

### Statistics and bootstrap resampling
Once the data were assigned to TN:TP windows, we randomly sampled them 300 times per TN:TP window (13800 iterations in total) using a hierarchical bootstrap procedure[60]. In short, we randomly sampled with replacement at the lake level, then randomly sampled one observation (either a single growing season average or a single 5-year SMA) from each lake without replacement until the number of samples equaled the number of lakes in the corresponding TN:TP window. For each random sample, we calculated generalised linear models (GLM, glm function, stats package in R)[56]. We used Chla as the dependent variable and built models using either TN (TN - Chla models), TP (TP - Chla models), or both TN and TP in additive (TN + TP - Chla) models. In the GLMs, we used the gamma distribution as the link function for the 5-year SMA and the normal distribution of Chla as the link function for SMA residuals (see SI for detailed descriptions of the SMA, TN:TP windowing and bootstrapping approaches). We kept only those iterations where all GLMs converged. Of the 13800 possible iterations (46 TN:TP windows times 300 iterations per window), the models converged for 9194 iterations of the 5-year SMAs and for 13800 iterations of the SMA residuals.

From the iterations where all GLMs converged, we extracted $R^2$ (as pseudo $R^2 = 1$ - model deviance/zero deviance), Akaike's information criterion (AIC), model slopes and intercepts (only $R^2$ and slopes are shown in the main manuscript, please see the SI for the other extracted variables). We used the delta AIC between models of the 5-year SMA to determine whether additive or interactive GLMs of TN and TP had higher model quality than those using only TN or TP as explanatory variables (see SI for model selection procedures). Model improvement was observed with additive TN + TP - Chla models, but no further improvement was obtained by including an interaction term. Therefore, we only show results for the TN - Chla, TP - Chla, and TN + TP - Chla models. To assess model improvement in terms of $R^2$ for the 5-year SMA compared to SMA residuals, we compared the distribution mean and 95% highest density interval (HDI)[61].

### Software and packages
All data was analysed in R, version 4.3.2 and partially (bootstrap procedure and generalized linear models) in Julia, version 1.9.0, using code developed by the authors. Within the data analysis code, Julia packages DataFrames (version 1.6.1), FLoops (version 0.2.1), DataFramesMeta (version 0.14.1), GLM (version 1.9.0) and Statistics were used; as well as the R package collection tidyverse (version 2.0.0).

Information about the number of lakes with average depth <6 m was extracted from the HydroLAKES dataset using QGIS Desktop (version 3.34).

The manuscript was written in Quarto Markdown (version 1.88.0) within the VSCodium IDE (version 1.79.0), and using R (version 4.3.2). Within the manuscript code, plots were generated with the ggplot2 (version 3.4.4), ggpubr (version 0.6.0), ggtext (version 0.1.2), and patchwork package (version 1.1.3).

### Reporting summary
Further information on research design is available in the Nature Portfolio Reporting Summary linked to this article.

## Data availability

This study uses open-access data from Filazzola et al.[55] ([https://doi.org/10.5063/F1RV0M1S](https://doi.org/10.5063/F1RV0M1S)), the LAGOS-NE database[58] ([https://lagoslakes.org/lagos-ne/](https://lagoslakes.org/lagos-ne/)), the Danish Overladevandsdatabasen ([https://odaforalle.au.dk/](https://odaforalle.au.dk/)) and from the HydroLAKES database[22] ([https://www.hydrosheds.org/products/hydrolakes](https://www.hydrosheds.org/products/hydrolakes)). All data used and generated in this study are also available here: [https://git.ufz.de/graeber/long-term-nutrient-chla-links-shallow-lakes](https://git.ufz.de/graeber/long-term-nutrient-chla-links-shallow-lakes).

## Code availability

Links to used open-access software and all code developed for this study are available here: [https://git.ufz.de/graeber/long-term-nutrient-chla-links-shallow-lakes](https://git.ufz.de/graeber/long-term-nutrient-chla-links-shallow-lakes). All code developed for this study is published under the BSD-3-Clause License (allowing open access and free software usage with full recognition of the original copyright).

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

## Acknowledgements

The work of D.G., D.B. and T.S. was funded by the Fourth Period of Programme-oriented Funding, Helmholtz Association of German Research Centres, Research Field Earth and Environment. M.J.M. was funded by Estonian Research Council grant PRG1954. E.J. was supported by the TÜBITAK program BIDEB2232 (project 118C250). M.S. was supported by Poul Due Jensens Fond (Grundfos Foundation).

## Author contributions

The study was initially conceived by D.G. and T.A.D., who also developed the statistical approach. The methodology and manuscript text was further developed by D.G. with contributions from T.A.D, D.B., T.S., M.S., E.J. and M.J.M on sampling methods, and implications of ecosystem-level nutrient depletion, algal physiology and ecology, and nutrient biogeochemistry for the study approach and interpretation; with further inputs from T.A.D. and D.B. from lake and landscape management perspectives; and contributions from T.A.D, M.J.M., T.S., D.B., E.J., M.S. and T.L.L. on technical aspects, and during the writing phase.

## Funding

## Competing interests

The authors declare no competing interests.
