## [Peer Review File · Nature Communications]

Consistent stoichiometric long-term relationships between nutrients and chlorophyll-a across shallow lakesREVIEWER COMMENTS

Reviewer #1 (Remarks to the Author):

This paper explores how stoichiometric relationships between total nitrogen and total phosphorus underlie chlorophyll a-nutrient relationships in shallow lakes (< 6 m) using a global database of lake water quality. Briefly, major findings include (1) there is a strong stoichiometric relationship underlying chlorophyll-nutrient relationships once short-term variation is removed, (2) this relationship held across different lake types (for lakes < 6 m) and regions, (3) findings for total phosphorus supported TN:TP thresholds commonly used to indicate P-limitation (TN:TP > 30), and (4) findings for total nitrogen did not support TN:TP ratios used to indicate N-limitation (TN:TP < 20) with relationships potentially being influenced by denitrification and high dissolved organic nitrogen pools at high TN:TP.

I think that this is an interesting paper that will be of interest to the readership of Nature Communications, and therefore fits the scope of the journal. Previous research studying empirical relationships between chlorophyll and limiting nutrients has often speculated that breakpoints or thresholds in the relationships (non-linearities) was caused by shifting nutrient limitation, which I think this paper has nicely demonstrated. The sole use of TN:TP ratios to indicate nutrient limitation without considering nutrient bioavailability is flawed, and this paper improves on that approach. Additionally, analyses are performed on a large dataset of lakes, and I applaud the authors for making their code available.

While the underlying approach and hypotheses are intuitive, my biggest critique of the paper is that the execution of the statistical approach used was more difficult to comprehend (as likely evidenced by the long Supplementary Information file), and this may limit the readership or citations of this paper. After fully reading the Supplementary Information file, the approach used made more sense to me, but this wasn't apparent on initial reading of the paper.

Below, I provide general and mostly specific comments to help improve the overall quality of the manuscript.

General comments:

1. The modeling approach used in this paper requires a very detailed description in the Supplementary Information, including a lengthy section describing simulations to demonstrate that the Simple Moving Average (SMA) approach can successfully decompose trends from short-term noise (variation; see Section 4). The Supplementary Information also contains justifications for the selection of the appropriate SMA model and regression model. While I think that the information was helpful, the lengthy descriptions also demonstrated that the approach is rather complex and not intuitive. I realize that this is a short format paper, but it might be helpful to re-evaluate if some of the information in the Supplementary Information should be in the main text.
2. I think that the structure of the manuscript could be streamlined a bit more, largely by removing the sub-headings between presentation of results. For example, the description of long-term results using 5-year SMAs (lines 87-9) is very general, and does not describe the relationships between chlorophyll and total phosphorus or total nitrogen, including whether or not these support hypotheses (yes for phosphorus, no for nitrogen). These results are not described until the following two sections (lines 146-51 for phosphorus; lines 155-9 for nitrogen). The same applies to results for the residuals (lines 89-90).
3. The only criterion used for including data in analyses was that the lakes were < 6 m depth, which was used to focus on shallow, polymictic lakes (lines 248-50). I believe that there were likely still thermally-stratified lakes included in the dataset, which begs the question of why was the entire dataset not used? Would the 10% of observations that came from deeper lakes be enough to mask relationships? I don't believe that previous studies exploring empirical relationships between chlorophyll and nutrients have excluded deeper lakes, which confounds comparisons.
4. The commentary regarding alternative pathways for N fate in high TN:TP lakes (lines 184-91) seems a little speculative. This was presented as a hypothesis, but it seemed like there was a lot of discussion regarding it for the short format of this paper.

Specific comments:

1. Line 21: Add citation for statement regarding harmful algal blooms (similar to construction for fish kills and methane).
2. Lines 31-2: At this point, it is unclear why is this important. Have shallow lakes been under-represented in the past, or were deeper lakes excluded because they have unique relationships? I think that this should be written as "data from all shallow lakes (here defined as average depth < 6m)..."
3. Line 40: Responses of what exactly?
4. Lines 56-8: Could also include differences in composition, cellular chlorophyll content, and accessory pigments.
5. Lines 99-101: I find the reference to the extinction debt model a bit confusing. Recommend deleting.
6. Lines 113-16: To me, it is not quite clear how to interpret this comparison. If the HDI for short-term variation is half that of the HDI for SMAs, does that mean that it is highly variable? It would be helpful to provide some context here.
7. Lines 120-1: Why would this just affect short-term variation? What about long-term shifts in phytoplankton communities resulting from eutrophication (e.g., cyanobacteria blooms) or increased cellular chlorophyll quotas due to light limitation?
8. Lines 136-8: Do you mean in regards to decomposing short-term fluctuations from long-term trends? There are numerous studies using Mann-Kendall tests to identify long-term trends in water quality parameters despite short-term variability. Please clarify the intended meaning.
9. Lines 174-6: Was a regression of nitrate on TN performed? This would give an indication of whether or not nitrate is driving high TN concentrations. Previous work from agricultural regions has shown that nitrate can drive high TN concentrations in lakes (Filstrup and Downing 2017).
10. Line 280: Should the window width for TN:TP be 2, not 3? Interval 0.1 seems really small
Lines 281-2.

References:

Christopher T. Filstrup & John A. Downing (2017) Relationship of chlorophyll to phosphorus and nitrogen in nutrient-rich lakes, *Inland Waters*, 7:4, 385-400, DOI: 10.1080/20442041.2017.1375176

Reviewer #2 (Remarks to the Author):

Graeber et al described the relationships between nutrients and chlorophyll-a across lakes with publicly available long-term data. They found that strong relationships consistently existed between nutrients and chlorophyll a for 5-year simple moving averages along a gradient of total nitrogen to total phosphorus ratios. The topic is interesting in general, and the novelty of this study is that the authors applied a big field-observation data to support the thresholds of nitrogen phosphorus deficiency. But this relationship has well reported in previous studies from field and laboratory experiments, and the applied statistical analyses are problematic due to the non-independence among the used measurements. I am sorry I could not be more positive in this case. I showed my major concerns for this study below.

Non-dependence among studied variables prevents further strong statistic supports of the key findings. The authors had very few variables playing around in this study, including Chla, TN, TP, SRP and NO₃-N. As the authors stated in the introduction, the measurements of TN and TP and phytoplankton biomass are not independent, and the relationships are often tautologous. This limitation is getting worse when the authors attempted to relate the nutrient-Chla relationships to the TN:TP ratios as the same nutrients were used more than one time. For instance, the relationships shown in Figure 4 are clearly not independent, and for instance, it is the relationship between chla and TP shown in Figure 4D, rather than their ratios between TN. I think the authors were describing the results which could not well supported by their data or statistical analyses.

The study is descriptive in general and lacks in-depth mechanisms behind the findings. The authors applied the data sets to test the thresholds of nutrients, which is consistent in general to

previous studies. However, few mechanisms could be revealed from the pure field observations, and the authors provided the potential explanations of chl-a-TN ratio changes by denitrification and dissolved proteins. I may not agree well these descriptive hypotheses could explain well explain the findings without solid data or experiments. I also do not agree that the authors could provide solid evidence to well explain these findings, which thus leaves the study descriptive in general without inspiring insights behind the well-known topic.

The treatments in statistical analyses are subjective and more explanations or analyses are needed. For instance, why was the 5 years average selected? What are the differences among the averages of different year windows? Why was the depth of 6 meters applied? Did the authors test other water depth thresholds for the generality of the findings? Why were the additive models selected for TN and TP as I am not well persuaded that these two variables could be simply treated in this way? How were the thresholds in TN:TP ratios found? Did the authors applied proper statistical analyses to reveal these thresholds?

Point-by-point response, revision 1: Consistent stoichiometric
long-term relationships between nutrients and chlorophyll-a
across lakes

Daniel Graeber^{1*} Mark J. McCarthy² Tom Shatwell³
Dietrich Borchardt¹ Erik Jeppesen^{4,5,6,7,8} Martin Søndergaard^{4,5}
Torben L. Lauridsen^{4,5} Thomas A. Davidson⁴

**General remarks**

We thank the reviewers for their insightful comments, which we address in detail below. All changes
in the text are marked by purple color and the respective manuscript lines are mentioned for each
change.

We also uploaded a new version of the manuscript and SI markdown files to the online repository.

**Reviewer 1**

**Comment 1**

*"I think that this is an interesting paper that will be of interest to the readership of Nature Communi-*
*cations, and therefore fits the scope of the journal. Previous research studying empirical relationships*
*between chlorophyll and limiting nutrients has often speculated that breakpoints or thresholds in the*
*relationships (non-linearities) was caused by shifting nutrient limitation, which I think this paper has*
*nicey demonstrated. The sole use of TN:TP ratios to indicate nutrient limitation without considering*
*nutrient bioavailability is flawed, and this paper improves on that approach. Additionally, analyses are*
*performed on a large dataset of lakes, and I applaud the authors for making their code available.*

*While the underlying approach and hypotheses are intuitive, my biggest critique of the paper is that*
*the execution of the statistical approach used was more difficult to comprehend (as likely evidenced by*
*the long Supplementary Information file), and this may limit the readership or citations of this paper.*
*After fully reading the Supplementary Information file, the approach used made more sense to me, but*

*this wasn't apparent on initial reading of the paper."*

**Reply**

We thank the reviewer for this positive comment. We improved readability by several measures, please
see our replies to the comments below.

**Comment 2**

*"1. The modeling approach used in this paper requires a very detailed description in the Supplemen-*
*tary Information, including a lengthy section describing simulations to demonstrate that the Simple*
*Moving Average (SMA) approach can successfully decompose trends from short-term noise (variation;*
*see Section 4). The Supplementary Information also contains justifications for the selection of the*
*appropriate SMA model and regression model. While I think that the information was helpful, the*
*lengthy descriptions also demonstrated that the approach is rather complex and not intuitive. I realize*
*that this is a short format paper, but it might be helpful to re-evaluate if some of the information in*
*the Supplementary Information should be in the main text."*

**Reply**

We are aware that the used statistical approach is non-standard, thus we included a depiction of
this approach in Figure 1 in the original submission. We now completely revised the figure caption,
shortening and straightening it, by excluding less important information and moving other information
to the main text. With that we hope to make our statistical approach easier to understand (see changes
in lines 72 - 78, line 79, and in caption of Figure 1).

Furthermore, the long SI addresses a lot of aspects of analyses which were supporting the approach
of the paper. Here, based also on comment 4 from reviewer, our selection of the length of the running
means was not clear. We added a better description of this approach in lines 291 - 294 of the revised
manuscript.

**Comment 3**

*"2. I think that the structure of the manuscript could be streamlined a bit more, largely by removing the*
*sub-headings between presentation of results. For example, the description of long-term results using*
*5-year SMAs (lines 87-9) is very general, and does not describe the relationships between chlorophyll*
*and total phosphorus or total nitrogen, including whether or not these support hypotheses (yes for*
*phosphorus, no for nitrogen). These results are not described until the following two sections (lines*
*146-51 for phosphorus; lines 155-9 for nitrogen). The same applies to results for the residuals (lines*
*89-90)."*

**Reply**

The first part of the results discussion is dedicated to the differences between short-term and long-
term variability and not to the specifics of the links between TN or TP and Chla. To accomodate the
comment of the reviewer to improve the flow of the manuscript, we moved the N and P discussion up,
and the discussion on short and long-term variability down. We have not marked those changes, as we
did not change content, only moved paragraphs around (and because we would have had to color the
entire part of the manuscript). Here we also reduced the number of subtitles to improve the reading
flow (all changed subtitles have been marked in bold plus purple color). We also shortened the section
on N (here all changes have been marked, see also reply to comment on N discussion below, lines 137
65 - 167) to improve the flow. Finally, with that we could also avoid mentioning the same results twice,
which further improves the flow.

**Comment 4**

*“3. The only criterion used for including data in analyses was that the lakes were < 6 m depth, which*
*was used to focus on shallow, polymictic lakes (lines 248-50). I believe that there were likely still*
*thermally-stratified lakes included in the dataset, which begs the question of why was the entire dataset*
*not used? Would the 10% of observations that came from deeper lakes be enough to mask relationships?*
*I don’t believe that previous studies exploring empirical relationships between chlorophyll and nutrients*
*have excluded deeper lakes, which confounds comparisons.”*

**Reply**

We chose 6 m because water samples from lakes with larger average depth have a lower probability
to represent the entire water column. Here¹ has shown that 6 m is a critical depth at which the
probability of stratification is 0.5. Thus shallower lakes are more likely mixed than stratified. Since
we know that the US data from the LAGOS-NE dataset is epilimnetic data, while the Danish data
from the Overfladevandsdatabasen integrates the entire water column, removing this depth cut-off
would more likely result in comparing different parts of lake water column. Furthermore, we do not
know the exact sampling strategy for the remaining data in the global dataset, and using the depth
cut-off makes sure that this data can compare well to the other data. This has been described in the
Methods section of the original article. We moved some text into the main text and added further
information in the Methods to make this clear (lines 97 - 99, lines 267 - 273).

We now also conduct the same analysis with data from all lakes without depth cutoff and present
the results in the SI (SI lines 284 - 302). Based on the statistical results, the approach seems highly
promising for all lake types, irrespective of lake depth. Specifically the R² patterns change for the

TP - Chla models, yet are otherwise similar to the data with depth cut-off, and the slope results are
highly comparable to the data with depth cut-off.

Still, we keep the depth cut-off, since the dataset is dominated by shallow lakes. Thus, if different
patterns would be the case for deeper lakes, we may not be able to see it and, from that deduct
false conclusions on deeper lakes. Furthermore, the lower number of deeper lakes precludes us from a
separate analysis of those. Finally, one would need to make sure that the water samples also do well
represent the entire water column, something which we cannot be certain of, as we already discuss in
the previous paragraph.

**Comment 5**

*“4. The commentary regarding alternative pathways for N fate in high TN:TP lakes (lines 184-91)*
*seems a little speculative. This was presented as a hypothesis, but it seemed like there was a lot of*
*discussion regarding it for the short format of this paper.”*

**Reply**

We find this discussion important. Some of author team have a biogeochemical perspective and
for those co-authors, the potentially overlooked importance of DON for lake N budgets is the most
important outcome of the paper. Still we recognize that this part is too long and we feel that is hard
to read in parts. Thus we shortened and revised it to make it not stand out too much (lines 148 - 167)

**Specific comments**

1. Line 21: Add citation for statement regarding harmful algal blooms (similar to construction for
fish kills and methane).

• Reply: Citation has been added (line 21)

2. Lines 31-2: At this point, it is unclear why is this important. Have shallow lakes been under-
represented in the past, or were deeper lakes excluded because they have unique relationships? I
think that this should be written as “data from all shallow lakes (here defined as average depth
< 6m)...”

• Reply: We agree with the reviewer, and changed the text accordingly (lines 31 - 32)

3. Line 40: Responses of what exactly?

• Reply: We meant variability of responses to changes in TN or TP concentrations at different
TN : TP. We corrected that (lines 39 - 41).

- 4. Lines 56-8: Could also include differences in composition, cellular chlorophyll content, and
accessory pigments.
- • Reply: We agree and argue with this also in the discussion of the results. We now shortly
mention this also here (lines 58 - 61).
- 5. Lines 99-101: I find the reference to the extinction debt model a bit confusing. Recommend
deleting.
- • Reply: Here, the model is an example for a delayed response of a species to a disturbance. We
agree that by itself the mentioning of the model is confusing. We now revise the text to the fact
that a disturbance effect (eg. change in nutrient input) may be delayed over long time scales,
mimicking the true impact when looking at short time scale data (lines 193 - 195)
- 6. Lines 113-16: To me, it is not quite clear how to interpret this comparison. If the HDI for
short-term variation is half that of the HDI for SMAs, does that mean that it is highly variable?
It would be helpful to provide some context here.
- • Reply: Well spotted. The reviewer is absolutely correct that the remaining variability is not
nearly as high as the one gathered by the long-term data. This was misrepresented in the text.
We revised this to make the difference clearer (lines 203 - 205).
- 7. Lines 120-1: Why would this just affect short-term variation? What about long-term shifts
in phytoplankton communities resulting from eutrophication (e.g., cyanobacteria blooms) or
increased cellular chlorophyll quotas due to light limitation?
- • Reply: We agree that this, in theory, might be important. Yet, statistically, this seems not
important in the long-term (here multi-year), as our models do not require any of those factors
to explain the Chla response to TN or TP with high R^2 . Specifically, variable Chla quotas
would occur due to changes in composition or cell-level changes. This would have increased
the variability of responses to TN or TP, something which would be visible in the range of
uncertainty of R^2 (the blue scatter around the yellow lines), which we found well constrained (in
contrast to the short term R^2 scatter, Fig. 2). Thus, from our perspective this is more important
to discuss for the short-term data.
- 8. Lines 136-8: Do you mean in regards to decomposing short-term fluctuations from long-term
trends? There are numerous studies using Mann-Kendall tests to identify long-term trends in
water quality parameters despite short-term variability. Please clarify the intended meaning.
- • Reply: Yet, in our opinion, studies fail to see that eutrophication is a long-term process. To our
knowledge no study clearly addresses this, and the same we find to be true in ecosystems outside

of lakes. This is what this paragraph is about. If the reviewer has specific studies in mind which
contrast with our viewpoint, we would be happy to know about them.

9. Lines 174-6: Was a regression of nitrate on TN performed? This would give an indication
of whether or not nitrate is driving high TN concentrations. Previous work from agricultural
regions has shown that nitrate can drive high TN concentrations in lakes (Filstrup and Downing
2017).

• Reply: We find that the very high TN concentrations (approx > 3 mg / L) are correlated to very
high nitrate-N concentrations, yet up to this range the relationship between TN and nitrate-N
is highly variable. We attribute this to the potential importance of DON capturing a lot of the
TN (please also see our reply to reviewer comment 5). We now added plots on TN and TP
correlations to nitrate-N and SRP at the end of the SI (SI section 11 & 12). Furthermore, we
added the reference in the text in the discussion on the links between TN and nitrate-N (lines
142 - 143).

10. Line 280: Should the window width for TN:TP be 2, not 3? Interval 0.1 seems really small Lines
281-2.

• Reply: No the logarithmic TN:TP window width is indeed 3. A step of $TN : TP = 0.1$ in
logarithmic space equals a true change in $TN : TP$ of from eg. $TN : TP = 20$ to 22.3 but
from eg. $TN : TP = 100$ to $TN : TP = 109$. Therefore the window moves different lengths in
non-logarithmic space. This follows statistic probability density distribution of $TN : TP$, which
is always logarithmic². Here, a larger logarithmic step would move the window much further for
higher $TN : TP$.

Reviewer 2

Comment 1

*“Graeber et al described the relationships between nutrients and chlorophyll-a across lakes with publicly*
*available long-term data. They found that strong relationships consistently existed between nutrients*
*and chlorophyll a for 5-year simple moving averages along a gradient of total nitrogen to total phos-*
*phorus ratios. The topic is interesting in general, and the novelty of this study is that the authors*
*applied a big field-observation data to support the thresholds of nitrogen phosphorus deficiency. But*
*this relationship has well reported in previous studies from field and laboratory experiments, and the*
*applied statistical analyses are problematic due to the non-independence among the used measurements.*
*I am sorry I could not be more positive in this case. I showed my major concerns for this study below.”*

**Reply**

We thank the reviewer for the comment. We show below in detail that the non-independence is largely
not the issue the reviewer proposes. However, the reviewer kindly remarked the non-independence
issue in figure 4 which we overlooked. We corrected this. Please see the reply to comment 2 for details.
Furthermore, as also stated in the same reply below, we use non-independence of total nutrients and
Chla as a tool, as it only should occur if a nutrient is depleted relative to other nutrients. Please see
also details below in the reply to comment 2.

We disagree with the reviewer on the missing novelty on the study. No previous study has reported
a global, robust, ubiquitous, stoichiometric deficiency pattern as our study does. Here, the reviewer
does not provide literature to support his/ her claim. Hence, we give the most important literature
below and compare it to our study:

1. Guildford and Hecky 2000³ is a kind of standard paper on global stoichiometric thresholds of
eutrophication, cited in a multitude of more recent literature, e.g.^{4,5}. In Guildford and Hecky
2000³, the authors find similar thresholds for nine study locations from the open ocean to lakes by
using a mix of field and laboratory studies. They have three locations with $TN : TP > 50$ (Lake
Superior, eight other large and small Ontario lakes) and two locations with $TN : TP < 20$ (Lake
Victoria and the Arctic), and use field and laboratory measurements combined with thresholds
of phytoplankton N and P growth deficiency based on earlier studies to determine N and P
limitation. Therefore, their dataset contains of data from a limited number of locations and
makes ecosystem level assumptions based on laboratory measurements of phytoplankton with
the notion that such indirect evidence that phytoplankton characteristics might point towards
ecosystem N or P deficiency. This is a good start for a search for global stoichiometric deficiency
patterns of eutrophication and, indeed, we find similar patterns for 159 lakes based on purely
field-based measurements, and across continents, irrespective of climate, land use, lake size,
catchment characteristics, with only average lake depth < 6 m as limitation (see also our reply
to comment 4 by reviewer 1 for details on that), and based on a rigorous bootstrap resampling
to find any uncertainty in our statistical models. Thus, our study provides direct empirical
evidence that the N and P deficiency proposed for a few locations in Guildford and Hecky 2000³
are in fact relevant in lake ecosystems with relevance for shallow lakes which make up more than
80% of all lakes globally.

2. A second milestone for our study is Moon et al. 2021⁴. Here, the authors use a US snapshot
sampling dataset from more than 1000 lakes with one sample per lake and also find similar
patterns as in Guildford and Hecky (2000)³, yet with high uncertainty and unclear indication
of co-depletion of N and P at intermediate $TN : TP$ ratios. Here, we clearly show the need

for the right temporal perspective with multi-year sampling and long-term means to detect the
statistical Chla response to changed nutrient stoichiometry with high confidence. Furthermore
we use another statistical approach than Moon et al. 2021⁴ with which we can show the co-
depletion and importance of both N and P at intermediate TN : TP ratios. Finally, the relatively
recent Moon et al. 2021⁴, cites no other relevant literature for large scale, global stoichiometric
patterns in lakes which would negate the novelty of our study.

3. A set of two studies, using statistical means to putatively detect N or P limitation based on TN
and TP are the papers of Dolman et al.^{6,7}. However, these studies do not take into account the
issue of Chla - TP tautology (see reply to Comment 2). Something that Dolman et al. do well
is to employ piecewise regression in acknowledgement of different links of TN or TP to Chla at
different N : P ratios. Yet, the authors cannot assess whether N and P co-depletion exists for
any of the lakes and assume strict Liebig-style limitation based on which it characterizes lakes
by month (although the authors themselves are aware that this does likely not exist in natural
communities based on their discussion). Furthermore, they do have considerable variability in
their data, requiring them to log transform algal biomass and nutrient concentrations similar
to Moon et al.⁴, and despite that, they do not reach the same strong, robust link as we do
with using 5-year SMAs. Thus, they fail to assess or recognize the importance of the correct
temporal perspective on eutrophication, an assessment which is central to our study. (We have
not mentioned their approach in our introduction, this has been changed in the revision (lines
44 - 45).).

In short, our study is novel due to (i) its global perspective, (ii) is assessment of the correct temporal
perspective to constrain the links between TN, TP and Chla, and (iii) its unique quantitative and
bootstrap-based statistics with strong emphasis of assessment of result robustness. All patterns we
find emerge from the data and were not induced by pre-existing assumptions. With that approach, our
study is the first to reveal an ecosystem-scale, globally ubiquitous stoichiometric relationship between
239 N, P and Chla for shallow lakes.

**Comment 2**

*“Non-dependence among studied variables prevents further strong statistic supports of the key findings.*
*The authors had very few variables playing around in this study, including Chla, TN, TP, SRP and*
*NO₃-N. As the authors stated in the introduction, the measurements of TN and TP and phytoplankton*
*biomass are not independent, and the relationships are often tautologous. This limitation is getting*
*worse when the authors attempted to relate the nutrient-Chla relationships to the TN:TP ratios as the*
*same nutrients were used more than one time. For instance, the relationships shown in Figure 4 are*

*clearly not independent, and for instance, it is the relationship between chl_a and TP shown in Figure*
*4D, rather than their ratios between TN. I think the authors were describing the results which could*
*not well supported by their data or statistical analyses.”*

**Reply**

We absolutely agree that in Figure 4, dependence of variables was introduced due to the calculation of
ratios between TN and the inorganic nutrient fractions. Hence, we now show absolute concentrations
of SRP and nitrate-N in Figure 4. The patterns are like the ones observed for the ratios before. Hence,
we come to very similar conclusions on the link between TN : TP and SRP or nitrate-N occurrence. We
include Chl_a as color coded variable now, which reveals that for nitrate-N the dichotomous behavior
of nitrate-N at high TN : TP persists and is reflected in very different Chl_a concentrations at this
range. In contrast, average Chl_a concentrations decrease with decreasing SRP with higher TN:TP (>
50) as expected when assuming Redfield ratio (molar N:P = 16) being ideal for phytoplankton growth,
and when running into P depletion at higher TN:TP. Please see the new figure 4, the changed figure
caption and the changed text linked to figure 4 in the revised manuscript (lines 128 - 131, lines 143 -
145)

The idea of the tautology is actually key to our approach, as we state in the introduction. As originally
stated in Lewis (2008)⁸:

*„Phosphorus and chlorophyll both are essential components of phytoplankton biomass. Therefore,*
*measurements of phosphorus and chlorophyll that are taken in a lake over the same span of time (e.g.,*
*the growing season) are not independent variables; there must always be a correlation between the*
*two variables, although the strength of the correlation will weaken if concentrations of phosphorus far*
*exceed the need of phytoplankton for phosphorus.”*

Thus, we agree with the assumption of some correlation, but, more importantly, utilize it for our
purpose to assess stoichiometric nutrient depletion of phytoplankton growth. Specifically, the correla-
tion must diminish when the concentration of a nutrient surpasses the requirement for phytoplankton
growth. This surpassing can occur either in terms of high concentrations or in terms of a “stoichio-
metric surpassing”, where a different nutrient, rather than the nutrient in question, becomes depleted
in concentration. Consequently, the nutrient in question suddenly becomes available in excess, which
leads to the weakening of the correlation between this nutrient and Chl_a. Our concept is strongly
supported by the evident systematic correlation patterns observed along the TN:TP gradient, as these
patterns strongly indicate such a “stoichiometric surpassing” for long-term averages.

**Comment 3**

*“The study is descriptive in general and lacks in-depth mechanisms behind the findings. The authors*
*applied the data sets to test the thresholds of nutrients, which is consistent in general to previous*
*studies. However, few mechanisms could be revealed from the pure field observations, and the authors*
*provided the potential explanations of chl-a-TN ratio changes by denitrification and dissolved proteins.*
*I may not agree well these descriptive hypotheses could explain well explain the findings without solid*
*data or experiments. I also do not agree that the authors could provide solid evidence to well explain*
*these findings, which thus leaves the study descriptive in general without inspiring insights behind the*
*well-known topic.”*

**Reply**

The study is strictly empirical, not descriptive, as it uses rigorous statistical testing. We have to
improved our explanations to clarify this (see our reply to comment 2 from reviewer 1). Any stoi-
chometric thresholds that emerge from the statistical analysis are a result of ubiquitously occurring
depletion patterns and are not in any way baked into the statistical approach.

In contrast to the notion of the reviewer, we feel that the study will be highly inspiring for further
research. It reveals the need to investigate improved stoichiometric management of shallow lakes
and their catchments, due to the globally ubiquitous stoichiometric patterns of lake eutrophication,
which is crucial for a planet of out of bounds in terms of its nitrogen and phosphorus cycles⁹. Our
long-term perspective on eutrophication is unique in contemporary freshwater sciences and proves
that long-term patterns in nutrient concentrations are decisive for describing eutrophication, patterns
which were expected in the 1980s by some of the most well known ecologists of that time^{10,11}. Thus,
our study provides unprecedented evidence for the importance of long-term eutrophication research
and lake monitoring. It will also be the jump off point for a host of experimental studies testing
the proposed mechanisms behind our global data analysis findings, including the proposed refractory
DON pool. Please also see our reply to your comment 1 and comment 1 of Reviewer 1 both of which
refute the claim of missing novelty of the study.

**Comment 4**

*“The treatments in statistical analyses are subjective and more explanations or analyses are needed.*
*For instance, why was the 5 years average selected? What are the differences among the averages of*
*different year windows? Why was the depth of 6 meters applied? Did the authors test other water*
*depth thresholds for the generality of the findings? Why were the additive models selected for TN and*
*TP as I am not well persuaded that these two variables could be simply treated in this way? How were*

*the thresholds in TN:TP ratios found? Did the authors applied proper statistical analyses to reveal*
*these thresholds?”*

**Reply**

The treatments were purely objective. In detail:

- 1. Length of averages: Please the see SI for an in-depth assessment for the ideal length of the
averages. This SI was already attached to the first submission and was likely overlooked by
the reviewer. To improve the understanding, we added text in lines 291 - 294 of the revised
manuscript.
- 2. “What are the differences among the averages of different year windows?” The authors are
uncertain about the meaning of this comment.
- 3. Lake depth limitation: We chose 6 m because water samples from lakes with larger average
depth have a lower probability to represent the entire water column. For details, see also our
reply to Comment 4 from Reviewer 1.
- 4. Additive models: An in-depth analysis of the ideal model type was done in the SI. This SI was
already attached to the first submission and was likely overlooked by the reviewer.
- 5. Thresholds: As stated above the changes in response of Chla to TN or TP along TN : TP
emerged from the data itself and are the result of the rigorous bootstrap analysis of TN, TN
and TN+TP - Chla models within TN : TP windows. This approach is stated in Figure 1, the
methods section and with further analyses in the SI. No pre-conceived thresholds were tested,
thus these are an inherent property of the data.

**References**

- 1. Kirillin, G. & Shatwell, T. Generalized scaling of seasonal thermal stratification in lakes. *Earth-*
*Science Reviews* **161**, 179–190 (2016).
- 2. Isles, P. D. F. The misuse of ratios in ecological stoichiometry. *Ecology* **101**, (2020).
- 3. Guildford, S. J. & Hecky, R. E. Total nitrogen, total phosphorus, and nutrient limitation in
lakes and oceans: Is there a common relationship? *Limnology and Oceanography* **45**, 1213–1223
(2000).
- 4. Moon, D. L., Scott, J. T. & Johnson, T. R. Stoichiometric imbalances complicate prediction
of phytoplankton biomass in U.S. Lakes: Implications for nutrient criteria. *Limnology and*
338 *Oceanography* **66**, 2967–2978 (2021).

- 5. Shatwell, T. & Köhler, J. Decreased nitrogen loading controls summer cyanobacterial blooms without promoting nitrogen-fixing taxa: Long-term response of a shallow lake. *Limnology and Oceanography* **64**, S166–S178 (2019).
- 6. Dolman, A. M. & Wiedner, C. Predicting phytoplankton biomass and estimating critical N:P ratios with piecewise models that conform to Liebig’s law of the minimum. *Freshwater Biology* **60**, 686–697 (2015).
- 7. Dolman, A. M., Mischke, U. & Wiedner, C. Lake-type-specific seasonal patterns of nutrient limitation in German lakes, with target nitrogen and phosphorus concentrations for good ecological status. *Freshwater Biology* **61**, 444–456 (2016).
- 8. Lewis, W. M. & Wurtsbaugh, W. A. Control of Lacustrine Phytoplankton by Nutrients: Erosion of the Phosphorus Paradigm. *International Review of Hydrobiology* **93**, 446–465 (2008).
- 9. Richardson, K. *et al.* Earth beyond six of nine planetary boundaries. *Science Advances* **9**, eadh2458 (2023).
- 10. Strayer, D. *et al.* Long-term ecological studies: An illustrated account of their design, operation, and importance to ecology. *Occasional Publication of The Institute of Ecosystem Sciences* **2**, 1–36 (1986).
- 11. Franklin, J. F. Importance and Justification of Long-Term Studies in Ecology. in *Long-Term Studies in Ecology: Approaches and Alternatives* (ed. Likens, G. E.) 3–19 (Springer, 1989).
doi:10.1007/978-1-4615-7358-6_1.

REVIEWERS' COMMENTS

Reviewer #1 (Remarks to the Author):

This paper explores how stoichiometric relationships between total nitrogen and total phosphorus underlie chlorophyll a-nutrient relationships in shallow lakes using a global database of lake water quality. This paper is a revised version of an originally submitted paper that was reviewed by two reviewers. In the revised version, I think that the authors have adequately and comprehensively addressed all reviewer critiques to the best of their abilities. Therefore, I think that this is an interesting paper that will be of interest to the readership of Nature Communications, and therefore fits the scope of the journal.

Below, I comment on several of the major critiques that were previously raised by the two reviewers.

1. Study novelty: I agree that the study is novel and nicely frames relationships between primary producer biomass and resource availability with a stoichiometric framework for shallow lakes. While chlorophyll-nutrient relationships are well-documented in lakes, shifting nutrient limitation status is often speculated to be the cause of these nonlinear relationships without robust statistical testing. Here, I think that this paper has accomplished that. In the detailed response, the authors have described how this paper builds on existing research, and therefore is novel.
2. Statistical approach: Both reviewers raised concerns about the complexity of the statistical approach and selection of key study parameters (i.e., depth cutoff, SMA window). In reading the revised version of the manuscript, the description of the statistical approach, including the revised Fig. 1 caption, seem clear to me. Further, the justification for the SMA window and depth cutoff are sound, with additional support for these parameters thoroughly described in the Supplementary Information.
3. Variable independence (tautology): The authors have carefully thought about the tautology argument and have adequately addressed it in the revised version of the manuscript. Chlorophyll-nutrient relationships are fundamental limnological relationships and provide the foundation for nutrient management strategies and the assessment of ecosystem health. Evaluating these relationships within a stoichiometric (nutrient limitation) framework and discovering robust patterns across numerous shallow lakes is informative.
4. Speculation of underlying mechanisms: Both reviewers raised concerns regarding speculation about the role of dissolved organic nitrogen (DON) and denitrification underlying chlorophyll-nutrient relationships in high TN:TP lakes with little supporting data. I think that this is my biggest concern with the revised paper. While a little speculation is appropriate in a paper, the original manuscript had lengthy commentary regarding this. I think that the shortened commentary in the revised paper lessens my concern.

Specific comments

1. Figure 4: Remove "The depicted R2 is based on Pearson's r of a linear regression." I think that this is text remaining from the original manuscript. If it is correct, then please add the r2 value to each plot.
2. Figure S2: Hard to visualize location of lakes. It would be helpful to add state boundaries.
3. SI Section 4.1.2: It would be helpful to indicate that k=15 was used to assess the accuracy of the SMA approach, even though k=5 was used in the paper. I was confused about this and thought it was a typo until I read SI Section 5.

**Reviewer 1**

**Comment 1**

*This paper explores how stoichiometric relationships between total nitrogen and total phosphorus un-*
*derlie chlorophyll a-nutrient relationships in shallow lakes using a global database of lake water quality.*
*This paper is a revised version of an originally submitted paper that was reviewed by two reviewers.*
*In the revised version, I think that the authors have adequately and comprehensively addressed all*
*reviewer critiques to the best of their abilities. Therefore, I think that this is an interesting paper*
*that will be of interest to the readership of Nature Communications, and therefore fits the scope of the*
*journal.*

*Below, I comment on several of the major critiques that were previously raised by the two reviewers.*

*1. Study novelty: I agree that the study is novel and nicely frames relationships between primary*
*producer biomass and resource availability with a stoichiometric framework for shallow lakes. While*
*chlorophyll-nutrient relationships are well-documented in lakes, shifting nutrient limitation status is*
*often speculated to be the cause of these nonlinear relationships without robust statistical testing. Here,*
*I think that this paper has accomplished that. In the detailed response, the authors have described how*
*this paper builds on existing research, and therefore is novel. 2. Statistical approach: Both reviewers*
*raised concerns about the complexity of the statistical approach and selection of key study parameters*
*(i.e., depth cutoff, SMA window). In reading the revised version of the manuscript, the description of*
*the statistical approach, including the revised Fig. 1 caption, seem clear to me. Further, the justifi-*
*cation for the SMA window and depth cutoff are sound, with additional support for these parameters*
*thoroughly described in the Supplementary Information. 3. Variable independence (tautology): The*
*authors have carefully thought about the tautology argument and have adequately addressed it in the*
*revised version of the manuscript. Chlorophyll-nutrient relationships are fundamental limnological*
*relationships and provide the foundation for nutrient management strategies and the assessment of*
*ecosystem health. Evaluating these relationships within a stoichiometric (nutrient limitation) frame-*
*work and discovering robust patterns across numerous shallow lakes is informative. 4. Speculation*
*of underlying mechanisms: Both reviewers raised concerns regarding speculation about the role of dis-*
*solved organic nitrogen (DON) and denitrification underlying chlorophyll-nutrient relationships in high*
*TN:TP lakes with little supporting data. I think that this is my biggest concern with the revised paper.*
*While a little speculation is appropriate in a paper, the original manuscript had lengthy commentary*
*regarding this. I think that the shortened commentary in the revised paper lessens my concern.*

**Reply**

We thank the reviewer for recognizing our efforts in improving the paper. The general comments do
not entail any further discussion or changes of the manuscript. Below, we reply to the remaining
specific comments raised by the reviewer.

**Specific comments**

1. *Comment: Figure 4: Remove “The depicted R^2 is based on Pearson’s r of a linear regression.”*
*I think that this is text remaining from the original manuscript. If it is correct, then please add*
*the r^2 value to each plot.*

• Reply: We indeed forgot to delete the text. We thank the reviewer for finding this issue.

2. *Comment: Figure S2: Hard to visualize location of lakes. It would be helpful to add state*
*boundaries.*

• Reply: We agree with the reviewer and added the US state boundaries.

3. *Comment: SI Section 4.1.2: It would be helpful to indicate that $k=15$ was used to assess the*
*accuracy of the SMA approach, even though $k=5$ was used in the paper. I was confused about*
*this and thought it was a typo until I read SI Section 5.*

• Reply: We are glad to see that our argumentation on choosing the SMA length was received
positively. Indeed a higher SMA length ($k = 15$) was best describing the simulated data. We
now inserted text to make clear that the k for the simulated data differed from the k of the real
lake data in section 5: “Be aware, this k is ideal for the simulated data and is different from the
ideal SMA length used for the observed lake data in main manuscript (which is $k = 5$). Below,
we describe in detail, how we arrive at the $k = 5$ for the lake data (Section 5).”